# Microscope-Based Augmented Reality with Intraoperative Computed Tomography-Based Navigation for Resection of Skull Base Meningiomas in Consecutive Series of 39 Patients

**DOI:** 10.3390/cancers14092302

**Published:** 2022-05-06

**Authors:** Mirza Pojskić, Miriam H. A. Bopp, Benjamin Saβ, Barbara Carl, Christopher Nimsky

**Affiliations:** 1Department of Neurosurgery, University of Marburg, 35037 Marburg, Germany; bauermi@med.uni-marburg.de (M.H.A.B.); sassb@med.uni-marburg.de (B.S.); barbara.carl@helios-gesundheit.de (B.C.); nimsky@med.uni-marburg.de (C.N.); 2Marburg Center for Mind, Brain and Behavior (CMBB), 35032 Marburg, Germany; 3Department of Neurosurgery, Helios Dr. Horst Schmidt Kliniken, 65199 Wiesbaden, Germany

**Keywords:** augmented reality, skull base surgery, skull base meningioma, intraoperative computed tomography, neuronavigation

## Abstract

**Simple Summary:**

The aim of surgery for skull base meningiomas is maximal resection with minimal damage to the involved cranial nerves and cerebral vessels. Compared to non-skull base meningiomas, these lesions show a reduced rate of gross total resection (GTR). Therefore, the use of technologies for improved orientation in the surgical field, such as neuronavigation and augmented reality (AR), is of interest. We confirmed in a consecutive series of 39 patients who underwent surgery for skull base meningiomas that automatic registration with intraoperative computed tomography (iCT) showed high registration accuracy and that microscope-based AR largely facilitated the resection by increasing surgical precision and providing improved intraoperative orientation by visualizing the tumor and the critical neurovascular structures in the operative microscope. No injuries to critical neurovascular structures occurred. There were 26 patients (66.6%) who underwent GTR. Additionally, 33 out of 35 patients who lived to follow-up could ambulate.

**Abstract:**

Background: The aim of surgery for skull base meningiomas is maximal resection with minimal damage to the involved cranial nerves and cerebral vessels; thus, implementation of technologies for improved orientation in the surgical field, such as neuronavigation and augmented reality (AR), is of interest. Methods: Included in the study were 39 consecutive patients (13 male, 26 female, mean age 64.08 ± 13.5 years) who underwent surgery for skull base meningiomas using microscope-based AR and automatic patient registration using intraoperative computed tomography (iCT). Results: Most common were olfactory meningiomas (6), cavernous sinus (6) and clinoidal (6) meningiomas, meningiomas of the medial (5) and lateral (5) sphenoid wing and meningiomas of the sphenoidal plane (5), followed by suprasellar (4), falcine (1) and middle fossa (1) meningiomas. There were 26 patients (66.6%) who underwent gross total resection (GTR) of the meningioma. Automatic registration applying iCT resulted in high accuracy (target registration error, 0.82 ± 0.37 mm). The effective radiation dose of the registration iCT scans was 0.58 ± 1.05 mSv. AR facilitated orientation in the resection of skull base meningiomas with encasement of cerebral vessels and compression of the optic chiasm, as well as in reoperations, increasing surgeon comfort. No injuries to critical neurovascular structures occurred. Out of 35 patients who lived to follow-up, 33 could ambulate at their last presentation. Conclusion: A microscope-based AR facilitates surgical orientation for resection of skull base meningiomas. Registration accuracy is very high using automatic registration with intraoperative imaging.

## 1. Introduction

In order to minimize the morbidity rate in patients with skull base tumors, integration of preoperative image-based segmentation with three-dimensional (3D) reconstruction of critical neurovascular structures with navigation was proposed [1]. Initial patient registration accuracy is a main factor influencing the overall navigation accuracy, and intraoperative imaging with intraoperative CT (iCT) provides user-independent patient registration [2,3,4]. Microscope-based augmented reality (AR) facilitated by the use of modern microscopes with integrated head-up displays (HUDs) provides visualization of 3D colored objects of interest in real time, which provides additional information and enables improved orientation in the surgical field [5,6]. Application of AR has been integrated into skull base surgery. Microscope-based AR has shown benefits in transsphenoidal surgery for pituitary adenomas and Rathke cysts, especially in cases with anatomic variants and in reoperations [7]. Further applications of AR in skull base surgery include endoscopic-assisted endonasal midline skull base pathologies in adults [8] and children [9], and resections of lesions of the lateral skull base in otorhinolaryngology [10].

Use of AR in resections of meningiomas has been previously described, with a total of 25 cases in the literature, but only 3 of them have been skull base lesions [11,12,13]. A recent case report demonstrated the use of AR for optimization of exposure in clinoid meningioma [13]. Skull base meningiomas are surgically challenging tumors due to the anatomy of the skull base and the proximity of neurovascular structures [14,15]. Thus, the aim of surgery for skull base meningiomas is maximal resection with minimal damage to the involved neurovascular structures [1]. Skull base lesions show a reduced rate of gross total resections compared to non-skull base meningiomas [16]; therefore, implementation of technologies for improved orientation in the surgical field, such as neuronavigation and AR, are needed, as especially cases of recurrent tumors and reoperations are associated with high morbidity and complication rates [15]. To our knowledge, this is the first study to evaluate the use of microscope-based AR for resection in a series of skull base meningiomas. 

## 2. Materials and Methods

Thirty-nine consecutive patients (thirteen male, twenty-six female, mean age 64.08 ± 13.50 years) who underwent surgery for skull base meningiomas at our department by a single surgeon (C.N.) between September 2016 and January 2022 using microscope-based AR and automatic patient registration using iCT were included in the study. The definition of skull base meningiomas was based on Al-Mefty et al. [17]. Indications for surgery included the presence of skull base lesions with neurological deficits, compression of neurovascular structures due to mass effect and presence of lesions primarily suspected of being brain metastases in patients with systemic cancer disease. Written informed consent was provided by all participants and their family members. Ethical approval for prospective archiving of all relevant clinical and technical data with no need for further approval for retrospective analysis was obtained. 

The extent of resection was determined on the postoperative MRI 3 months following surgery and was deemed as a gross total resection (GTR) with dural coagulation or removal of adjacent dura and bone (GTR, Simpson Grade I with complete excision including dura and bone, or Grade II with reliable coagulation of the dura attachment [18], subtotal resection (STR) with complete excision but insufficient dural coagulation or bone excision (Simpson Grade III, using a modified classification by Combs et al. with non-visible rest on MRI [19], partial resection (PR, Simpson Grade IV, incomplete excision, but remainder visible on MRI) or open biopsy (Simpson Grade V).

### 2.1. General Setup

Automatic registration based on iCT without user interaction was performed using a 32-slice movable CT scanner (AIRO, Brainlab, Munich, Germany). Details about the setup, intraoperative CT scanning protocols and conversion factors for determining the effective radiation dose (ED) have been previously described [4]. 

Segmentation of the tumor was usually performed in the T1-weighted post-contrast MRI modality, manually or using autosegmentation with an anatomical mapping element (Brainlab). Brainlab was used to delineate the optic nerves, chiasm, ventricles, pituitary gland or brain stem with unique colors being assigned to each object, and with additional manual segmentation for correction of the automatic segmentation and for segmentation of tumor and risk structures. For segmentation of the vascular risk structures, contrast-enhanced T1-weighted magnetic resonance imaging (MRI), time-of-flight (TOF) MRI angiography or computed tomography (CT) angiography was used and then rigidly registered. Rigid registration was performed by an image fusion element (Brainlab, Munich, Germany). This 3D data set formed the basis for patient registration [7] (Figure 1). 

Head fixation following narcosis and positioning for the appropriate approach was established by applying a radiolucent carbon Doro head clamp. Metallic fixation pins were placed outside the area of the intraoperative scan. The reference array with four reflective markers was attached to the head clamp [4]. Registration scanning was performed following the rotation of the operating table 90° prior to skin incision. Alternatively, registration scanning was performed following craniotomy. The transfer of images was performed automatically by the navigation system, followed by automatic registration with the application of images from the registration iCT scan. Three fiducial markers placed in the scan were of the skin and which were not used for the process of registration, were used to check the accuracy of the automatic registration. For each of the three fiducials, the individual target registration error (TRE) could be measured by placing the tip of the pointer in the divot of the fiducial [4].

Rigid registration of the preoperative plan with the registration CT scan was performed by image fusion software, Brainlab elements (Brainlab, Munich, Germany). Accuracy of the fusion was checked by a spy-glass feature, with visualization of the preoperative images in an insert of the intraoperative CT, which was moved around for detection of registration failure in all views with appropriate anatomical landmarks (Figure 1). 

### 2.2. Augmented Reality 

For AR support, the HUDs of the operating microscopes (Pentero 900 (Zeiss, Oberkochen, Germany) or Kinevo 900 (Zeiss, Oberkochen, Germany)) were used. Tracking of the microscope was performed by a registration array attached to the microscope. Checking the calibration of AR was performed by centering the microscope above the divot of the registration array, as well as additional markers. In this way, the optical outline and the AR visualization of the reference array could be adjusted and 3D objects could be visualized in semitransparent, solid or outlined mode using the AR display and the microscope application allowed for visualization of these objects in different modes on the microscope video [7].

## 3. Results

General characteristics of the patients are summarized in Table 1. Twenty-six patients (66.6%) underwent GTR of the meningioma. The mean surgery time was 300.4 ± 134.1 min. Most common were olfactory meningiomas (*n* = 6), meningiomas of the cavernous sinus (*n* = 6) and clinoidal meningiomas (*n* = 6), meningiomas of the medial (*n* = 5) and lateral (*n* = 5) sphenoid wing and meningiomas of the sphenoidal plane (*n* = 5), followed by suprasellar meningiomas (*n* = 4), falcine meningioma of anterior fossa (*n* = 1) and middle and posterior fossa/petrous meningioma (*n* = 1). Sixteen tumors (41%) had a volume more than 10 cm^3^ with encasement and compression of the neurovascular structures. 

Registration accuracy: Registration scanning was performed prior to skin incision in patients no. 1–29 (*n* = 28) and following craniotomy in patients no. 29–39 (*n* = 11). Automatic registration applying intraoperative CT in patients 1–29 resulted in high accuracy (target registration error, 0.82 ± 0.37mm). The mean dose-length product (DLP) of the scout and registration scan was 243.16 ± 392.34 mGy*cm. Patients 1–16 and 29–39 received a registration scan following the scout scan, whereas patients 17–28 received a registration scan without a prior scout scan. The effective radiation dose of the registration CT scans (patients 17–28) and scout scans (patients 1–16 and 29–39) was 0.58 ± 1.05 mSv. 

In 13 patients (patients numbered 1, 2, 5, 6, 7, 8, 9, 10, 11, 15, 16, 17 and 20) a control iCT scan for exclusion of perioperative complications and control of the extent of resection was performed. Repeat iCT scanning was used for updating navigation. 

Augmented reality: The major indications to select a patient for surgery using the AR application were invasive tumors with encasement of the carotid and medial cerebral arteries, all tumors with close relation to the optic chiasm, giant tumors (tumor volume > 10 cm^3^) or recurrent tumors. Accuracy of patient registration and microscope registration were the two dependent variables for clinical AR accuracy. Microscope registration accuracy was checked by applying the AR visualization of the reference array outline. Landmark checks were successfully performed, apart from checking the target registration error (TRE), which confirmed high accuracy and excluded errors due to potential shift (Figure 2).

AR improved orientation in the surgical field for all patients as it reliably visualized the structures of interest and closely matched them to the visualized objects and the visible tumor outline. This was particularly useful in giant tumors with encasement and displacement of the cerebral arteries and compression of the optic chiasm (patients 4, 6, 8, 9, 10, 12, 13, 14, 18, 20, 22, 27, 31, 32, 33 and 36) as well as in cases of recurrent tumors (patients 24 and 35). No injuries to critical neurovascular structures occurred. The tumor was visualized in all patients, followed by vessels of interest (27 patients or 62.3%) and the optic chiasm and optic nerves in 22 patients, or 56.4%. Individual objects or HUDs could be switched off on the preference of the surgeon, in the case of AR information overflow. If this was the case, further AR support was provided by the standard navigation display and AR display on the video screens, which allowed the assisting staff to monitor the surgery, thereby serving as an educational tool.

### Illustrative Cases

Case 1: Patient no. 28 was a 77-year-old female patient with right clinoidal meningioma who experienced visual field deficits and visual deterioration. A complete resection of the tumor was performed via right fronto-temporal craniotomy. AR support facilitated the course of the resection with prompt localization of the segmented carotid and cerebral arteries, as well as the optic chiasm, providing surgical precision throughout the procedure. Figure 3 demonstrates navigation and AR support at the beginning of the tumor resection and Figure 4 shows the microscopic view following the gross total resection of the tumor. The patient recovered fully and was neurologically intact. Operative video has been added to Appendix A.

Case 2: A 38-year-old female patient with giant medial sphenoid wing meningioma on the left side presented with vertigo (patient number 9). GTR was performed using iCT-based navigation registration and microscope-based AR. Figure 5 demonstrates the intraoperative view throughout the resection and Figure 6 shows preoperative and postoperative MRI imaging. The patient recovered fully and had no neurological deficits. Operative video has been added to the Appendix A. 

Case 3: A 67-year-old female patient (patient number 19) with visual deterioration and bitemporal hemianopsia. MRI showed a suprasellar meningioma which was resected via right pterional approach. Figure 7 demonstrates the visualization of the structures in the AR throughout the surgery. 

Case 4: The patient was a 55-year-old male with multiple intracranial meningiomas (patient no. 31). He underwent two surgeries and radiotherapy for lateral sphenoid wing meningioma on the left side. Due to tumor progression, he underwent subtotal resection (Simpson Grade III) via left fronto-temporal approach. Histology indicated an atypical meningioma WHO II°, and postoperative particle radiation therapy (carbon protons) was performed. The operative video demonstrates the surgical resection of the tumor using microscope-based AR (Appendix A). 

Case 5: A 72-year-old patient presented in a comatose state following several days of mental symptoms (patient number 36). He underwent GTR for WHO I° olfactory meningioma via bifrontal approach. Following surgery, the patient experienced prolonged weaning followed by tracheotomy. Furthermore, a hydrocephalus developed, and a subduro-peritoneal shunt was implanted. The patient recovered to an extent that allowed him to walk; however, assistance in everyday life is necessary.

Clinical outcome: An overview of clinical characteristics of the patients is provided in Table 2. 

Only one patient deteriorated postoperatively, with a new CN III palsy (patient no. 7). CN III palsy in this patient improved during the postoperative course. Eight patients improved following surgery and thirty patients remained unchanged. Perioperative surgical complications occurred in six patients (15.4%) and included cerebrospinal fluid (CSF) leaks which required surgical interventions (patients no. 4 and 23), wound healing deficits which prompted wound revision (patient no. 9), as well as postoperative hydrocephalus with implantations of a ventriculoperitoneal (patient no. 28) and subduro-peritoneal shunt (patient no. 36). Patient no. 20, with giant olfactory meningioma, experienced a massive pulmonary embolism five days following surgery, which caused asystolia, cardiopulmonary reanimation and diffuse brain swelling due to hypoxia. She underwent bifrontal craniectomy and subsequent ventriculoperitoneal shunt implantation. This patient, unfortunately, remained having an apallic syndrome. 

Further non-operative complications included heart failure due to asystolia with cardiopulmonary reanimation (patient no. 11), a pulmonary embolism which required cardiopulmonary reanimation (patient no. 14), pneumonia (patient no. 16) and sinus venous thrombosis with a pulmonary embolism (patient no. 29). These four patients recovered fully. 

The mean follow-up time was 26.7 ± 21 months. Four patients died before follow-up. Patient no. 8 died eleven days after surgery due to septic shock caused by pancolitis. Patient no. 18 died three months following surgery due to a pulmonary embolism. Patients no. 26 and 34 underwent surgery for skull base lesions suspected of brain metastases; however, the histology has shown a diagnosis of WHO I° meningioma. Patient no. 26 died 7 months following surgery due to progression of a renal cell carcinoma, and patient no. 34, died 3 months following surgery due to progression of liver failure caused by metastases of malignant melanoma. 

Patients numbered 1–4, 16, 13, 14, 16, 22, 30, 31, 32, 35 and 39 underwent postoperative particle radiotherapy. Indications for radiotherapy were Simpson Grade III, IV and V resections, tumors with invasion of the cavernous sinus or a WHO II° histology. The average time range from surgery to radiotherapy was 5.62 ± 1.94 months. Tumor recurrence and progression occurred in four patients (patients no. 2, 3, 16 and 31). Patient no. 2 experienced recurrence 60 months following surgery and was proposed surgical treatment, but refused further therapy. Patient no. 3 underwent two further surgeries due to the tumor’s progression (30 months and 55 months following primary surgery) and subsequent irradiation. Patient no. 16 was followed up with due to asymptomatic progression, which occurred 6 months following radiation therapy. Patient no. 31 experienced a tumor progress in the setting of meningiomatosis and WHO II° histology 16 months following surgery. This patient underwent a re-radiation for tumor progression. At the follow-up, 33 out of 35 patients could ambulate at their last presentation.

## 4. Discussion

A skull base location, WHO II° and higher pathology grades, together with a subtotal resection, are independent predictors of unfavorable outcomes in meningiomas [20]. Compared to non-skull base meningiomas, patients with skull base meningiomas have more frequent neurological deficits and impairment of these deficits following surgery, less GTR and shortened retreatment free survival, which does not affect overall survival [16]. As shown here, the use of neuronavigation and AR could assist in improved orientation in the surgical field and lead to less adverse advents and injury to neurovascular structures, possibly increasing the rate of the extent of resection. The issue of brain shift and the subsequent need for intraoperative image updating is less problematic in skull base lesions due to the small extent of intraoperative shift of the tumor and vessels because of their attachment to the bony structures of the skull base [1]. In our study, 67% of patients underwent GTR of the tumor. This is comparable to the current data in the literature on series of resections of skull base meningiomas. A study on 1148 consecutive patients demonstrated that GTRs are less frequent in skull base compared to non-skull base meningiomas (62% vs. 84%) [16]. However, skull base meningiomas include several types of tumors at different locations; thus, the extent of resection in conjunction with surgical and clinical outcomes may vary. Surgical management has further evolved from classical surgery to surgery with standard use of advanced imaging and neuronavigation, use of preoperative embolization in selected cases, radiotherapy and radiosurgery, as well as drug therapy, genetic profiling and oncologic management [21]. Treatment objectives consist of GTR wherever possible, with preservation of patients’ functions, quality of life and independence in daily living as the postoperative goal (Karnofsky performance score KPS > 70%). If GTR is not possible due to a high risk of endangering patients’ functions and quality of life, adjuvant treatment is mandatory [21]. Therefore, the optimal treatments for certain types of skull base meningiomas have evolved. It has become common in the clinical practice to perform subtotal or partial resection of the cavernous sinus meningiomas, following irradiation of the residuals [21]. The trend of aggressive resection of cavernous sinus and other skull base meningiomas has declined since the 1980s and 1990s, with contemporary series showing GTR rates of 53%, complication rates of 17.9% and 0.9% and adjuvant treatment rates of 22.2% [21]. A study by De Maio et al., in a series of 117 patients with skull base meningiomas, reported that 90.3% of patients had a KPS > 80% at follow-up [21]. 

Despite the evolution of different treatment options for skull base meningioma, the most possible, safest extent of resection still plays a central role in the therapy. AR can greatly enhance the safety of the resection, improving the orientation in the operative field and early identification of critical structures. Spatial accuracy of preoperative image segmentation with intraoperative data for skull base surgery has already been investigated [1]. The surgeon is not challenged to merge the neuronavigation data with the operative field by himself/herself, with AR performing this task instead, which is advantageous compared to traditional neuronavigation [6]. In the current literature only three cases of AR use for skull base meningiomas have been reported [11,12,13]. Use of microscope-based AR has shown to be reassuring, particularly for less experienced surgeons [1]. Whereas bony and neurovascular landmarks present reliable navigational tools for experienced surgeons, these landmarks are not fixed in all cases; thus, the use of 3D segmentation and AR can detect anatomical variations [1]. Neurovascular structures which are often invaded or displaced by skull base tumors can be visualized on MRIs, segmented and superimposed on the microscope in real-time fashion, which enables identification of these structures and is considered to be the decisive advantage in the resection of these lesions [1]. Although various devices have been used to achieve AR, including smartphones, tablets and head-mounted devices [22], microscope-based AR is currently the best way to apply AR due to an obligation to use the microscope for skull base surgery. 

When clear anatomical landmarks are lacking, for example, in cases of large destructive skull base tumors and reoperations, AR can decisively assist in comprehending the 3D surgical anatomy and improve the comfort, precision and intraoperative orientation of the surgeon [5]. Standards to assess the quality of AR visualization in neurosurgery are not well established [11] and thus far rely on the subjective impression of the surgeon. A recent study on utilization of AR in cranial surgery of 55 cases reported that 66.7% of surgeons found AR visualization helpful in individual cases with acceptable accuracy and depth information [11]. Furthermore, in cases of large tumors with encasement of the cerebral vessels, using AR during resection provides a good orientation and depth-in perception to the localization of the vessels inside the tumor, which reduces the risk of damaging these structures. Even if positional shift occurs, such as movement of the registration array, the size of the object remains displayed correctly. This facilitates the estimation of the extent of tumor resections [5]. 

Monitoring of the AR clinical accuracy can be achieved by evaluating intraoperatively anatomical landmarks that can be clearly located on the pre- or intraoperative images [7]. Using visible bony structures of the skull base that can be superimposed with a semitransparent display of a bony reconstruction of the CT data to adjust navigation, fine tuning of AR navigation can be performed [7]. The 3D display of AR objects provided intuitive depth perception and a very close match with the visible extent of the tumor, vessels and optic nerves, and its AR representation was observed. The possibility to switch on and off each object can eliminate the information overflow by AR objects [5]. AR usage in relation to microscope time varies between 44% (11) and 51% [5]. 

In order to rely on AR to perform surgery close to neurovascular risk structures, it is mandatory to have a high navigation accuracy. Patient registration using user-independent intraoperative imaging-based methods is highly recommended [7]. The main use of intraoperative CT imaging for cranial procedures has been found to be in skull base and neurovascular surgery, where resection is the most effective therapy for preservation of the recovery of visual function, especially in skull base tumors of orbital structures, paraorbital structures and optic nerves [23]. The use of neuronavigation was recommended earlier in skull base surgery as the normal anatomy is often distorted, leading to a higher risk of misidentification of anatomic landmarks and resulting in potential functional damage of neural structures [24]. In skull base surgery, the problem of brain shift does not pose a crucial obstacle for the use of AR and neuronavigation based on solid bony structures which can be used for navigation updates and checks. Due to integration of neuronavigation into the operative workflow, its use has become standard for skull base surgery in neuro- and ear, throat and nose (ENT) surgery [25]. 

Thus far, there are several studies that examined the role of neuronavigation and intraoperative imaging in the resection of skull base lesions [4,7,8,23,24,26,27,28]. Recent studies reported that use of navigation led to decreased recurrence rate, blood loss and length of hospitalization, and improved recurrence-free-survival and performance status in resections of meningiomas [29,30]. The main indication for iCT in cranial surgery is registration scanning [4]. An iCT-based neuronavigation allows referencing in the same patient position in which the surgery is performed. This leads to elimination of position-dependent brain shift and improves navigation accuracy [24]. A drawback of iCT-based automatic registration is that the patient is exposed to radiation [27]. Low-dose protocols allowed for significant reduction of the ED (effective dose). Thus, iCT can be routinely applied due to precise depiction of bony landmarks and high neuronavigation accuracy. Furthermore, a negligible brain shift in the region of the skull base and a short duration of examination even enables repeated iCT scans if needed, without significantly prolonging the surgery time [24]. Intraoperative control of resection with a second control scan as a standard procedure for all skull base operations is not necessary, yet it can be helpful in a highly selected cohort of patients where the extent of resection is affected by distorted anatomy and the involvement of osseous structures that need to be checked intraoperatively [24]. Recent studies on orbit-associated tumors, most of them being meningiomas, showed that intraoperative imaging caused the change in surgical approach and strategy, due to an unexpected residual tumor or additional tumor that was initially not visualized because of the overlay of osseous tumor parts [24]. In our study, control iCT scans were performed to exclude perioperative complications and for estimation of the extent of resection. However, no complications occurred and the surgical strategy was not changed according to the control iCT. 

Future directions may include surgical rehearsals with AR-templates for complex skull base cases. A recent video article on the resection of clinoid meningioma demonstrated the use of surgical rehearsal in virtual reality (VR) with an AR-template, where the author demonstrated how AR-enhanced navigation was used for the planning of the incision and soft-tissue exposure and for the guiding of the drilling of the sphenoid wing and the extradural clinoidectomy [13]. Use of VR and AR-templates according to an individualized patient’s anatomy could cause a paradigm shift where the surgeon at the surgery duplicates the plan which he/she has already rehearsed [13]. This application would also be an excellent educational tool for residents and less experienced surgeons and an important addition—but not a complete substitution—to traditional skull base surgery training in a cadaver laboratory. 

There are several limitations of this study. This is a single surgeon retrospective study, which makes its results and conclusions only partially reproducible. However, prospective studies with the use of AR, including its use in the educational setting with less experienced surgeons and the possibility of presurgical rehearsal with AR-templates, are needed to objectively evaluate the use of AR for resection of skull base meningiomas. In our study, follow-up time was too short to make adequate conclusions on the course of the disease, yet our primary goal was to demonstrate the use of AR for the surgical resection, whereas further follow-up is needed for assessment of the long-term outcomes. Since the AR was switched off in cases of information overflow, the microscopic-based AR did not always provide support throughout the entire procedure. There is no control group with patients who underwent resection without use of iCT-based navigation and AR. However, a control group of patients who would undergo surgery without optimal conditions is considered unethical and unpractical. 

## 5. Conclusions

A microscope-based AR facilitated surgical orientation for resection of skull base meningiomas. Registration accuracy was very high using automatic registration with intraoperative imaging. AR proved to be very useful in cases of large skull base meningiomas which encase cerebral vessels through early identification of risk structures and in cases of tumors which compressed the optic chiasm and optic nerves, as well as in reoperations. Enhanced understanding of 3D anatomy could be of potential use as an educational tool and assistance for less experienced surgeons. 

## Figures and Tables

**Figure 1 cancers-14-02302-f001:**
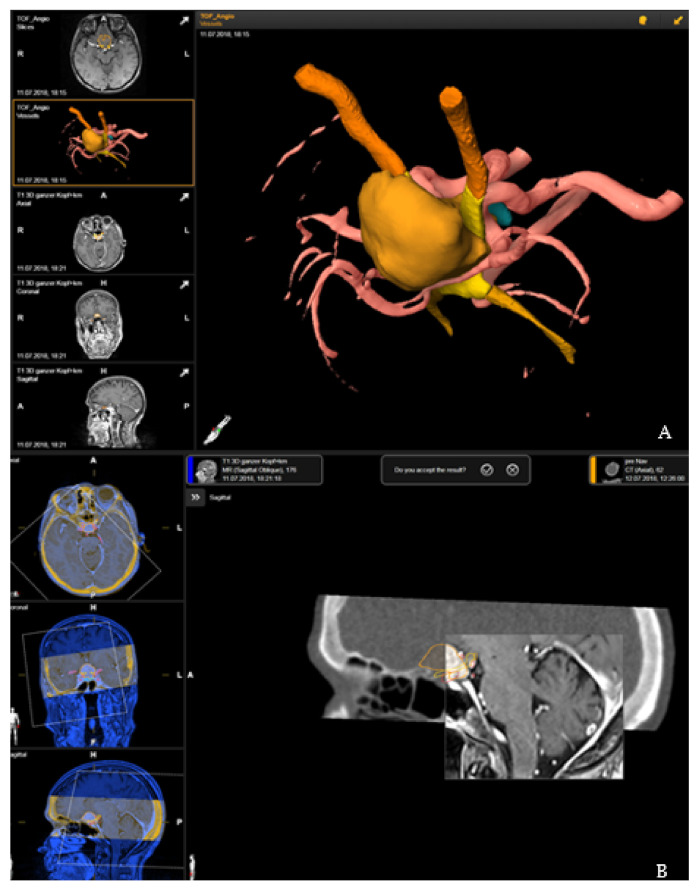
Preoperative planning and image fusion for 67-year-old female patient with suprasellar meningioma (patient no. 19). (**A**) Visualization of the tumor (ochre), optic nerves and tract (orange) and optic chiasms (yellow) as a 3D object following segmentation in T1-weighted post-contrast magnetic resonance imaging (MRI) and time-of-flight (TOF) MRI angiography. (**B**) Rigid fusion of computed tomography scan with T1-weighted MRI scan.

**Figure 2 cancers-14-02302-f002:**
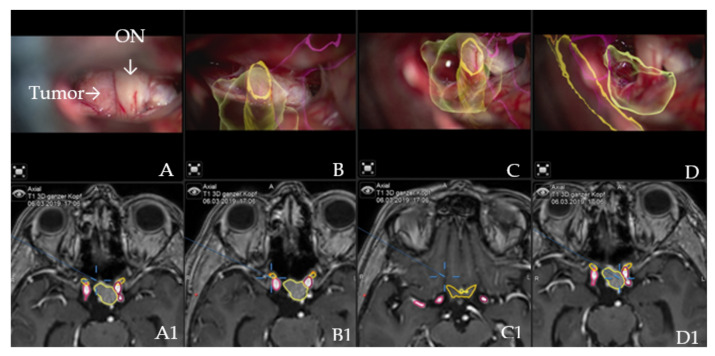
AR accuracy check. Patient no. 21 underwent a resection for suprasellar meningioma via right pterional approach. (**A**) Microscope video with focus on the tumor following exposure, with (**A1**) T1-weighted post-contrast MRI axial view of standard navigation display with segmented objects (tumor, optic chiasm and nerves in yellow and carotid arteries in violet). Focus of the microscope is shown as seen on the standard navigation display. (**B**) Microscope video with head-up display and 3-dimensional (3D) visualization of the segmented objects during the resection, with focus on right optic nerve, which shows high accuracy with the intraoperative situation. (**B1**) T1-weighted post-contrast MRI axial view of navigation, with focus on right optic nerve. (**C**) Microscope video with head-up display and 3D visualization of tumor outline, optic nerve and ipsilateral carotid artery following tumor resection with focus on the skull base with (**C1**) T1-weighted post-contrast MRI axial view of navigation. (**D**) Same as in C, microscope focus is on the contralateral carotid artery; segmented tumor outline and the course of the contralateral optic nerve are visualized with (**D1**) T1-weighted post-contrast MRI axial view of navigation, which shows focus of the microscope as seen on the standard navigation screen.

**Figure 3 cancers-14-02302-f003:**
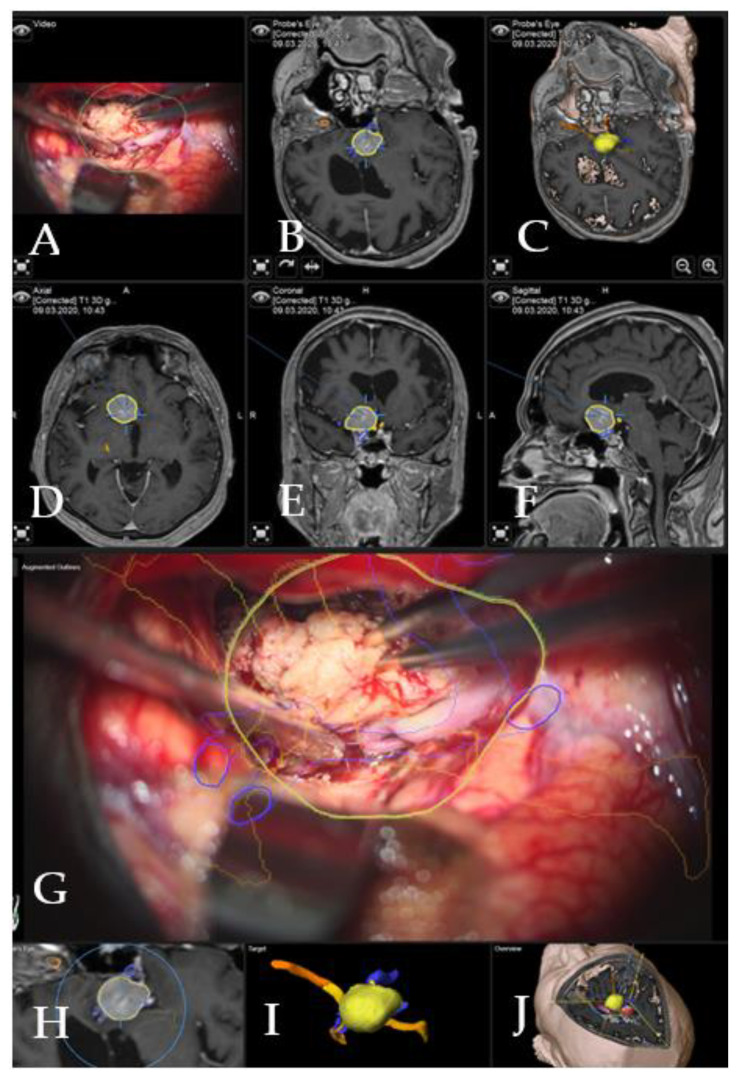
Navigation and AR support during surgery (patient no. 28). (**A**) Microscope video with head-up display and 3-dimensional (3D) visualization of the segmented objects (tumor in yellow, carotid and anterior cerebral arteries in blue, optic chiasm in yellow). (**B**,**C**) Probe’s eye view in 2D and 3D mode. Navigation display in (**D**) axial, (**E**) coronal and (**F**) sagittal view with the segmented objects (focus on the tumor following debulking). (**G**) AR display on video screen with the 3D outline of tumor, carotid arteries, optic nerves and chiasm. (**H**) Corresponding probe’s eye view. (**I**) Target view (tumor and further objects outside of the focus plane are visualized) and (**J**) video plane in relation to the 3D objects.

**Figure 4 cancers-14-02302-f004:**
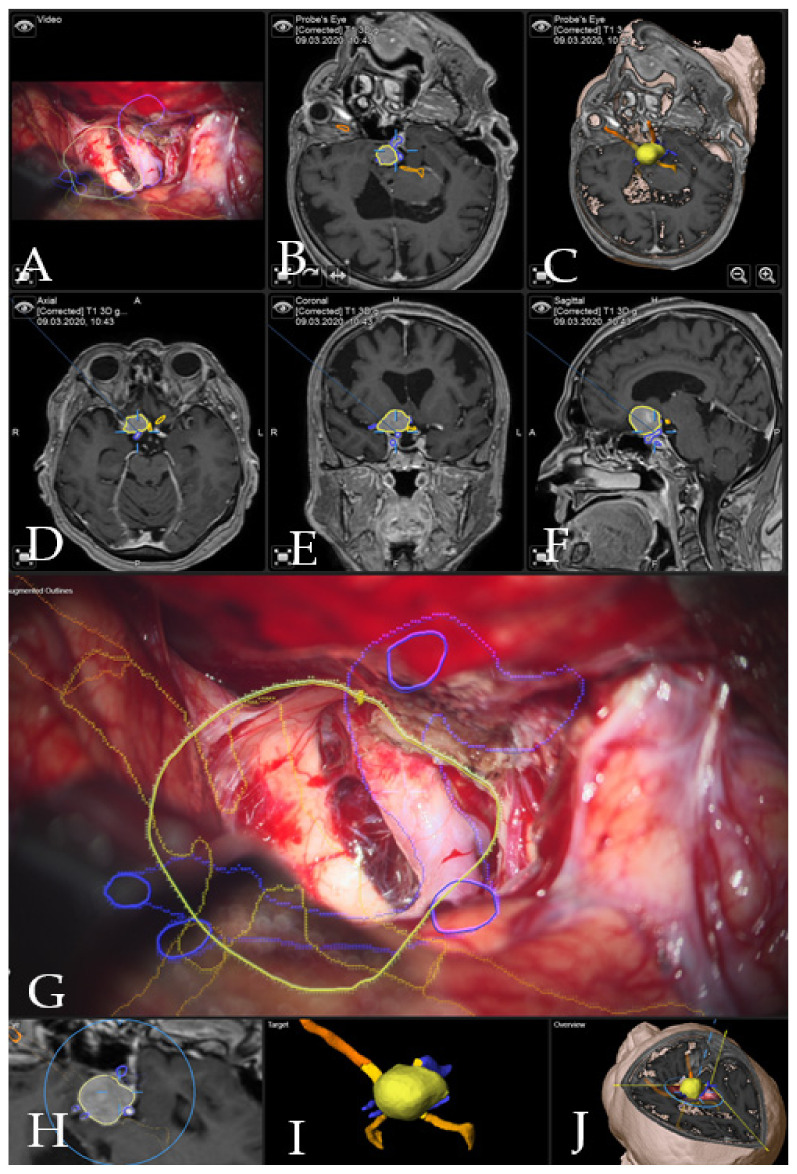
Navigation and AR support following complete resection of the tumor (patient no. 20). (**A**) Microscope video with head-up display and 3-dimensional (3D) visualization of the segmented objects (tumor in yellow, carotid and anterior cerebral arteries in blue, optic chiasm in yellow). (**B**,**C**) Probe’s eye view in 2-dimensional and 3D fashion. (**D**) Axial, (**E**) coronal and (**F**) sagittal views of standard navigation display with the segmented objects (focus on the sellar floor following complete resection). (**G**) AR display on video screen with the 3D outline of segmented structures. (**H**) Corresponding probe’s eye view. (**I**) Target view and (**J**) overview depicting the video plane in relation to the segmented 3D objects.

**Figure 5 cancers-14-02302-f005:**
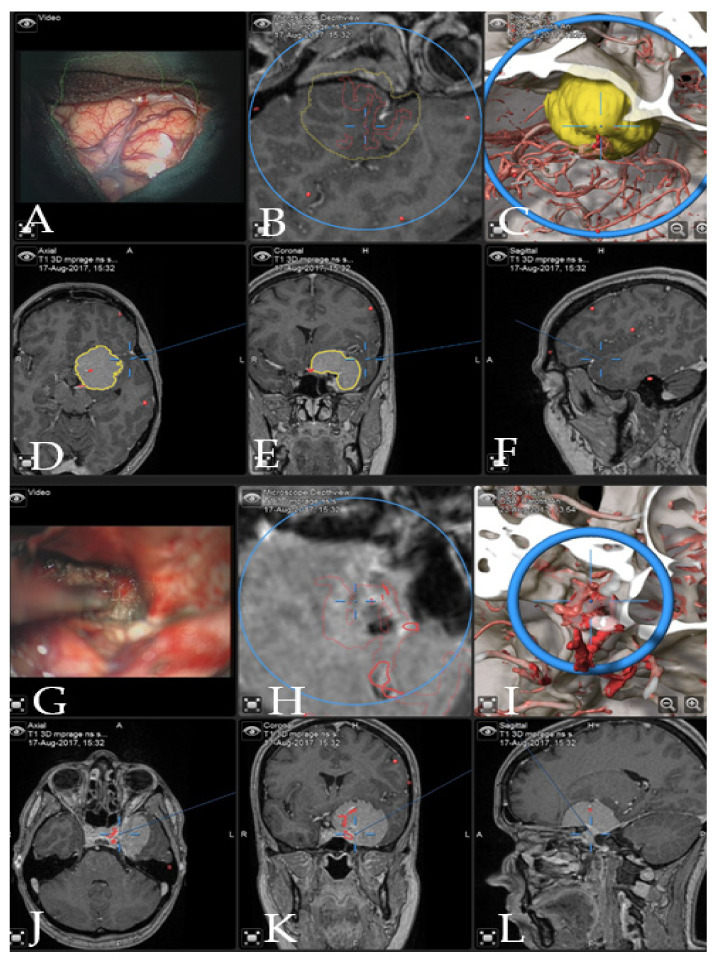
Navigation and AR support following complete resection of the tumor (patient no. 9). (**A**) Microscope video with head-up display and 3-dimensional (3D) visualization of the segmented objects (tumor in yellow, carotid, anterior and media cerebral arteries in red) following craniotomy. (**B**,**C**) Probe’s eye view in 2D and 3D fashion. (**D**) Axial, (**E**) coronal and (**F**) sagittal views of standard navigation display with the segmented objects. (**G**) Microscope video during the resection at the point where carotid artery in cavernous sinus is reached with (**H**,**I**) probe’s eye view in 2D and 3D fashion with AR objects on the screen. (**J**) Axial, (**K**) coronal and (**L**) sagittal views of standard navigation display with the segmented objects.

**Figure 6 cancers-14-02302-f006:**
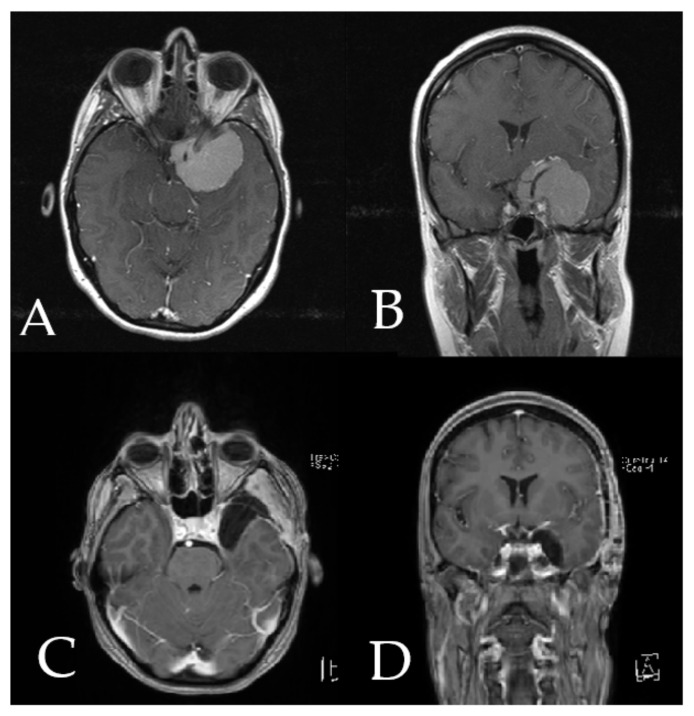
Preoperative axial (**A**) and coronal (**B**) T1-weighted post-contrast MRI of the head in patient number 9 shows a large medial sphenoid wing meningioma with encasement of carotid and cerebral medial artery. Postoperative axial (**C**) and coronal (**D**) postcontrast MRI of the head shows complete resection of the tumor.

**Figure 7 cancers-14-02302-f007:**
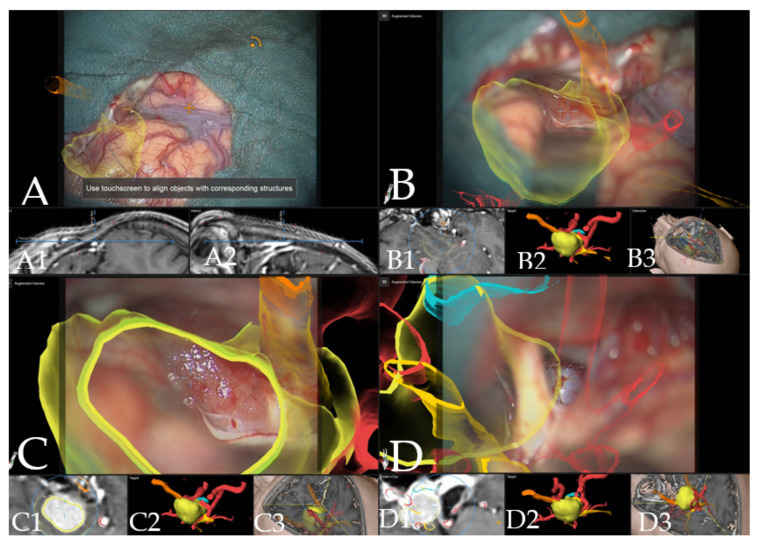
Navigation and augmented reality support during surgery (patient no. 19, same patient as in Figure 1). (**A**) Three-dimensional visualization of the segmented objects as seen through the microscope video with head up display (tumor in yellow, optic chiasm and optic nerves in orange), showing navigation update after craniotomy with focus on Sylvian fissure in (**A1**) axial and (**A2**) coronal view of the standard navigation. (**B**) Three-dimensional visualization of segmented objects (with vessels in red) following retraction of the frontal lobe with (**B1**) corresponding probe’s eye view, (**B2**) target view and (**B3**) overview depicting the video plane in relation to the segmented objects in 3D fashion. (**C**) Microscope video of tumor exposure with head-up display with (**C1**) corresponding probe’s eye view, (**C2**) target view and (**C3**) overview of the video plane. (**D**) AR display on video screen with the 3D outline of tumor, carotid arteries, optic nerves and chiasm with focus on the carotid artery, following complete resection of the tumor with (**D1**) probe’s eye view, (**D2**) target view and (**D3**) overview of the video plane.

**Table 1 cancers-14-02302-t001:** General characteristics of the cohort.

No.	Sex	Age (Years)	Location of the Tumor	Extent of Resection	Simpson Grade	Tumor Volume (cm^3^)	Visualized Objects in AR
1	m	57	Cavernous sinus/petroclival right	PR	IV	11	Tumor, Vessels
2	f	64	Clinoidal left	GTR	II	13.4	Tumor, Vessels, Chiasm, Optic Nerves, Optic Tracts, Ventricles
3	m	55	Cavernous sinus right	Open Biopsy	V	3.3	Tumor, Vessels, Chiasm, Optic Nerves, Optic Tracts
4	m	67	Anterior fossa/planum sphenoidale	GTR	II	53	Tumor
5	m	80	Olfactory meningioma	GTR	I	14.3	Tumor, Chiasm, Optic Nerves, Optic Tracts, Ventricles, Cerebrum
6	f	50	Middle and posterior fossa/petrous left	PR	IV	68.8	Tumor
7	m	65	Medial sphenoid wing meningioma left	GTR	II	15.8	Tumor, Vessels, Chiasm, Optic Nerves, Optic Tracts
8	f	89	Anterior fossa/planum sphenoidale	STR	IV	44.2	Tumor, Vessels, Chiasm, Optic Nerves, Optic Tracts
9	f	38	Medial sphenoid wing left	GTR	II	39.2	Tumor, Vessels
10	m	81	Lateral sphenoid wing right	GTR	II	64.6	Tumor, Vessels
11	f	76	Olfactory meningioma	GTR	I	10.7	Tumor, Chiasm, Optic Nerves, Optic Tracts
12	f	76	Olfactory meningioma	GTR	II	60.4	Tumor
13	m	48	Lateral sphenoid wing left	STR	III	98.2	Tumor, Vessels, Chiasm, Optic Nerves, Optic Tracts
14	f	558	Cavernous sinus left	PR	V	62.1	Tumor, Vessels, Chiasm, Optic Nerves, Optic Tracts
15	f	83	Clinoidal meningioma right	GTR	II	3.18	Tumor, Vessels, Chiasm, Optic Nerves, Optic Tracts
16	f	80	Planum sphenoidale	STR	IV	3.47	Tumor
17	m	46	Olfactory meningioma	GTR	II	5.91	Tumor
18	f	51	Anterior fossa/planum sphenoidale	GTR	II	53.4	Tumor, Vessels
19	f	67	Suprasellar meningioma	GTR	I	8.45	Tumor, Vessels, Chiasm, Optic Nerves, Optic Tracts, Pituitary Gland
20	f	50	Anterior fossa/falx	GTR	II	109	Tumor, Vessels
21	f	71	Suprasellar meningioma	GTR	II	1.32	Tumor, Vessels, Chiasm, Optic Nerves
22	m	82	Cavernous sinus right	STR	III	26.7	Tumor, Vessels, Chiasm, Optic Nerves, Optic Tracts
23	f	82	Suprasellar meningioma	GTR	II	6.05	Tumor, Vessels, Bone, Chiasm, Optic Nerves, Optic Tracts
24	m	71	Recurrent clinoidal meningioma right	STR	III	0.54	Tumor, Vessels, Bone, Chiasm, Optic Nerves, Optic Tracts, Ventricles, Cerebrum
25	f	60	Olfactory meningioma	GTR	I	1.66	Tumor
26	f	68	Medial sphenoid wing right	GTR	II	1.28	Tumor
27	f	54	Lateral sphenoid wing right	GTR	II	20.6	Tumor, Vessels, Chiasm, Optic Nerves, Optic Tracts
28	f	77	Clinoidal meningioma right	GTR	I	8.43	Tumor, Vessels, Chiasm, Optic Nerves, Optic Tracts
29	f	67	Cavernous sinus meningioma left	PR	V	11	Tumor, Vessels
30	f	71	Clinoidal meningioma right	GTR	I	3.34	Tumor, Vessels, Chiasm, Optic Nerves, Optic Tracts
31	m	55	Recurrent lateral sphenoid wing left	STR	IV	24.8	Tumor, Vessels, Chiasm, Optic Nerves, Optic Tracts
32	f	41	Lateral sphenoid wing right	STR	IV	29.8	Tumor, Vessels, Chiasm, Optic Nerves, Optic Tracts
33	m	64	Medial sphenoid wing right	GTR	II	28.2	Tumor, Vessels, Chiasm, Optic Nerves, Optic Tracts
34	f	70	Clinoidal meningioma right	GTR	I	2.92	Tumor, Chiasm, Optic Nerves, Optic Tracts
35	f	47	Recurrent medial sphenoid wing right	PR	IV	7.12	Tumor, Vessels
36	m	72	Olfactory meningioma	GTR	I	68.9	Tumor, Vessels
37	f	71	Anterior fossa/planum sphenoidale	GTR	II	1.4	Tumor
38	f	52	Cavernous sinus/temporobasal meningioma left	GTR	II	1.8	Tumor
39	f	45	Suprasellar meningioma right	GTR	II	1.91	Tumor, Vessels, Chiasm, Optic Nerves, Optic Tracts

**Table 2 cancers-14-02302-t002:** Clinical characteristics of the patient cohort.

No.	Symptoms	Neurological Deficits Prior to Surgery	Neurological Deficits Following Surgery
1	Incidental following mild TBI	None	None
2	Visual symptoms	Visual deterioration (Visus R 0.3, L 0.7)	Improved vision (Visus R 0.5, L 1.0)
3	CNVI palsy	CNVI palsy	CNVI palsy
4	Tremor	Tremor	None
5	Anosmia, headache	Anosmia	Anosmia
6	Headache	None	None
7	Seizure	None	CNIII palsy
8	Blindness right eye	Blindness right eye	Unchanged
9	Vertigo	None	None
10	Double vision	None	None
11	Headache	Anosmia	Anosmia
12	Gait difficulties, anosmia	Anosmia	Anosmia
13	Visual symptoms	Visual deterioration	None
14	Visual symptoms	CNIII palsy	CN III palsy
15	Visual symptoms	Visual deterioration (R 0.2, L 0.7)	Improved vision (R 0.2, L 0.8)
16	Visual symptoms	Visual deterioration (R 0.1, L 0.4)	Improved vision (R 0.2, L 0.3)
17	Seizure	None	None
18	Seizure	None	None
19	Visual symptoms	Visual det (R 0.7, L 0.8), bitemp. hemianopsia	Improved vision (R 0.7, L 0.9)
20	Hydrocephalus	None	None
21	Hypesthesia in right upper extremity	None	None
22	Seizure	None	None
23	CNIV palsy	CNIV palsy	CN IV palsy
24	Visual symptoms	Visual deterioration (L 0.16)	Improved vision (L 0.2)
25	Incidental finding	None	None
26	Screening due to renal carcinoma	None	None
27	Depression	None	None
28	Visual symptoms	Visual field deficits R	None
29	CN IV palsy	CN IV palsy	CN IV palsy
30	Depression	None	None
31	None- MRI follow-up following prior surgery	CN IV palsy, CN III palsy	CN IV and CN III palsy
32	Visual symptoms	Visual deterioration (R 0.6, L 1)	Improved vision (R 1.0)
33	Visual symptoms, dementia	Visual deterioration (R 0.2, L 0.05)	Unchanged vision (R 0.2, L 0.05)
34	Visual symptoms	Visual deterioration (R 0.7)	Unchanged vision (R 0.7)
35	None- MRI follow-up following prior surgery	None	None
36	Coma	Anosmia	Anosmia
37	Headache	None	None
38	Cognitive problems	None	None
39	Headache	None	None

Table legend: CN—cranial nerve, R—right eye, L—left eye, TBI—traumatic brain injury.

## Data Availability

The data in this study are available on request from the corresponding authors.

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
