# Peer review of "Microscope-Based Augmented Reality with Intraoperative Computed Tomography-Based Navigation for Resection of Skull Base Meningiomas in Consecutive Series of 39 Patients"

_cancers, 2022, doi:10.3390/cancers14092302_

Round 1

Reviewer 1 Report

Dear Authors,

Thank you for submitting your manuscript investigating usage of neuronavigation with combination of augmented reality for resection of skull base meningiomas. Study involves 39 consecutive patients from a single center. Design is a cohort without a control group, which is a weakness for structural and scientific objectivity, but the authors are well aware of the situation and carefully noted why there can't be a control group because of ethical considerations. Limitations of the study are clarified in accordance. Case series involves enough number of patients to make assumptions about benefits of augmented reality. Literature involves similar studies but with higher number of patients, data presentation, supplementary video files of surgical excision with augmented reality fusion images. The study has an overall high quality but needs some minor revisions and addressing of some unclear statements before publication.

1) Line 102 involves a sentence "For segmentation of the vascular risk structures, contrast-enhanced T1-weighted magnetic resonance imaging (MRI), time-of-flight (TOF) MRI angiography, or computed tomography (CT) angiography was used. These data sets were rigidly registered with a 3D image data set using the image fusion element (Brainlab, Munich, Germany) and formed the basis for patient registration (7) (Figure 1)." Is the reference number 7 is relevant here? Please check the accuracy.

2) Line 156 involves a sentence "The effective radiation dose of the registration CT scans (patients 17-28) with scout (patients 1-16 and 29-39) was 0.58 ± 1,05 mSv." This statement needs further clarification about the separation of the patients into two groups.

3) Information readily available from figures and tables does not need to be repeated in sections of the manuscript text. Results section can be shortened in terms of this suggestion. On the other hand, figure 7-9 have great visuals but the text is poor regarding the explanation of these figures.” This part does not apply to this particular manuscript. It was written for another reviewed study. I do apologize for this mix-up.

(By the way, I would like to add a humble opinion and a suggestion. Authors should consider adding “navigation” or “neuronavigation” to the list of the keywords to gain more visibility.)

4) Do all of these 39 patients' surgeries involved usage of augmented reality? What are the inclusion and exclusion criteria? The manuscript gives information about surgeon biased turning off the augmented reality feedback due to possibility of misorientaion. Could this also be a limitation of the study?

5) As a suggestion, clinical outcome subheading of the "Results" section should be supported by a table or a graph to facilitate understanding.

6) At line 167 "TRE" and at line 415 "ED" abbreviations are used but there are no definitions provided for both.

7) Figure 2 legend A is not clear. Please check the accuracy.

8) Figure 5 has mislettering of subfigures in the legend (second usage of B and C should be H and I respectively.). D, E and F subfigures are similar with J, K and L subfigures and their legends are the same.

9) In Figure 7, there is no B3 subfigure, yet there is in its legend.

10) Line 303 involves a sentence beginning with "Patient no. 2. died 7 months following surgery due to progression of the renal cell carcinoma,..." I assume the correct version will be patient 26.

11) There are several English language misusage in the manuscript. If not, English language editing should be performed preferably by a professional manuscript proofreading service.

These topics should be well covered. After addressing these issues and minor revision, manuscript would be eligible for publication.

Author Response

Response to Reviewer #1:

Point 1. Thank you for submitting your manuscript investigating usage of neuronavigation with combination of augmented reality for resection of skull base meningiomas. Study involves 39 consecutive patients from a single center. Design is a cohort without a control group, which is a weakness for structural and scientific objectivity, but the authors are well aware of the situation and carefully noted why there can't be a control group because of ethical considerations. Limitations of the study are clarified in accordance. Case series involves enough number of patients to make assumptions about benefits of augmented reality. Literature involves similar studies but with higher number of patients, data presentation, supplementary video files of surgical excision with augmented reality fusion images. The study has an overall high quality but needs some minor revisions and addressing of some unclear statements before publication.

Response 1. We appreciate very much compliments of Reviewer#1 on our study.

Point 2. 1) Line 102 involves a sentence "For segmentation of the vascular risk structures, contrast-enhanced T1-weighted magnetic resonance imaging (MRI), time-of-flight (TOF) MRI angiography, or computed tomography (CT) angiography was used. These data sets were rigidly registered with a 3D image data set using the image fusion element (Brainlab, Munich, Germany) and formed the basis for patient registration (7) (Figure 1)." Is the reference number 7 is relevant here? Please check the accuracy.

Response 2. We appreciate for opportunity to offer a clarification on this. Reference [7] is relevant here, since the segmentation of the vascular risk structures and rigid registration of these data sets with a 3D image data set using the image fusion element, which formed a basis for patient registration, is the method which our research group already described and published in an article on use of augmented reality in transsphenoidal surgery (Reference [7]. Carl B, Bopp M, Voellger B, Saß, B., Nimsky, C. Augmented Reality in Transsphenoidal Surgery. World Neurosurg. 2019;125:e873-e83.). (Page 3, Line 108-114)

Point 3. 2) Line 156 involves a sentence "The effective radiation dose of the registration CT scans (patients 17-28) with scout (patients 1-16 and 29-39) was 0.58 ± 1,05 mSv." This statement needs further clarification about the separation of the patients into two groups.

Response 3. We appreciate for opportunity to offer a clarification on this point. Patients 1-16 and 29-39 received a registration scan following scout scanning, which resulted in increased effective radiation dose due to the scout. Patients 17-28 received a registration scan without prior scout scan. Following first 16 cases, we have deemed scout scan as unnecessary for patients who received registration scan prior to craniotomy, so that the latter cases received a scan without scout. In patients 29-39, who received registration scan following craniotomy, scout scan was necessary to estimate the exact area covered by the registration scan. We have added this explanation accordingly (Page 7, Line 179-181).

Point 4. 3) Information readily available from figures and tables does not need to be repeated in sections of the manuscript text. Results section can be shortened in terms of this suggestion. On the other hand, figure 7-9 have great visuals but the text is poor regarding the explanation of these figures.” This part does not apply to this particular manuscript. It was written for another reviewed study. I do apologize for this mix-up. (By the way, I would like to add a humble opinion and a suggestion. Authors should consider adding “navigation” or “neuronavigation” to the list of the keywords to gain more visibility.)

Response 4. We appreciate Reviewer #1 suggestion for improvement. We have added “neuronavigation” to the list of keywords to gain more visibility (Page 1, Line 43)

Point 5. 4) Do all of these 39 patients' surgeries involved usage of augmented reality? What are the inclusion and exclusion criteria? The manuscript gives information about surgeon biased turning off the augmented reality feedback due to possibility of misorientation. Could this also be a limitation of the study?

Response 5. We appreciate for opportunity to offer a clarification on this. All 39 patients with skull base meningiomas underwent surgery with the use of augmented reality since the introduction of this technology in our department. The major indications to select a patient for surgery with the AR application were invasive tumors with encasement of the carotid and medial cerebral arteries, all tumors with close relation to optic chiasm, giant tumors (tumor volume > 10 cm3), and recurrent tu-mors (Page 2, Line 79-82). Exclusion criteria included patients who underwent surgery in sitting position due to the lack of possibility to perform intraoperative CT. Individual objects or head-up display could be switched off on the preference of the surgeon, in the case of information overflow. In this case, further augmented reality support was provided by the standard navigation display and AR display on the video screens. Since the AR was switched off in case of the information overflow, the microscopic-based AR did not always provide support throughout the entire procedure. We have included this aspect to the limitations part of the study (Page 20, Line 515-517).  

Point 6. 5) As a suggestion, clinical outcome subheading of the "Results" section should be supported by a table or a graph to facilitate understanding.

Response 6. We appreciate Reviewer #1 suggestion for improvement. We have added a Table which summarizes the main clinical findings in our cohort (Table 2) (Page 15, Line 313-320).

Point 7. 6) At line 167 "TRE" and at line 415 "ED" abbreviations are used but there are no definitions provided for both.

Response 7. We thank Reviewer #1 suggestion for improvement. We have added the explanation of the terms target registration error (TRE) and effective dose (ED) to Materials and Methods part of the manuscript accordingly (Page 3, Line 126-133). Three fiducial markers placed in the scan are of the skin and which are not used for the process of registration, were used to check the accuracy of the automatic registration. For each of the three fiducials individual target registration error (TRE) could be measured by placing the tip of the pointer in the divot of the fiducial. The intraoperative CT registration scanning protocols consisted of a lateral scout scan (10 mA, 120 kV) for slice alignment and the actual registration scan applying different protocols. The dose length product (DLP) refers to a phantom with a diameter of 16 cm. For an estimation of the effective radiation dose (ED), current ED/DLP conversion factors were estimated to be 2.4 μSv/mGy*cm for head scans (HudaW, Magill D, HeW(2011) CT effective dose per dose length product using ICRP 103 weighting factors. Med Phys 38:1261– 1265. https://doi.org/10.1118/1.3544350; Huda W, Ogden KM, Khorasani MR (2008) Converting dose length product to effective dose at CT. Radiology 248:995–1003. https://doi.org/10.1148/radiol.2483071964). Technique of estimation of TRE and estimation of ED has been previously reported and described in the research articles of our working group, which was cited accordingly (Page 3, Line 95-98). 

Point 8. 7) Figure 2 legend A is not clear. Please check the accuracy.

Response 8. We thank Reviewer #1 suggestion for improvement. We have added labels of tumor and optic nerve in the Figure 2A. Furthermore, we have changed the legend of the figure for clarification (Page 8, Line 198-209).

Point 9. 8) Figure 5 has mislettering of subfigures in the legend (second usage of B and C should be H and I respectively.). D, E and F subfigures are similar with J, K and L subfigures and their legends are the same.

Response 9. We thank Reviewer #1 suggestion for improvement. We have corrected the mislettering in the description of the figure (Page 12, Line 269). Subfigures and the legends of subfigures D, E and F as well as J, K and L are similar, since these subfigures depict the standard navigation display with different microscope focus on the different parts of the surgery (A-F on the beginning of surgery following craniotomy, and G-L following debulking and resection, when cavernous sinus has been reached). 

Point 10. 9) In Figure 7, there is no B3 subfigure, yet there is in its legend.

Response 10. We thank Reviewer #1 suggestion for improvement, we have added the subfigure B3 and formatted the figure (Page 14, Line 281).

Point 11. 10) Line 303 involves a sentence beginning with "Patient no. 2. died 7 months following surgery due to progression of the renal cell carcinoma,..." I assume the correct version will be patient 26.

Response 11.  We thank Reviewer #1 for this remark. We have changed the sentence accordingly (Page 16, Line 351-354).

Point 12. 11) There are several English language misusage in the manuscript. If not, English language editing should be performed preferably by a professional manuscript proofreading service.

Response 12. We thank Reviewer #1 suggestion for improvement. The manuscript underwent proofreading by an academic researcher, who published extensively in English language academic literature. English language editing was performed accordingly and we have added the acknowledgment. (Page 20, Line 544)

Point 13. These topics should be well covered. After addressing these issues and minor revision, manuscript would be eligible for publication.

Response 13. We thank Reviewer #1 for effort and detailed review of our study. We hope that our corrections and changes are adequate and sufficient to recommend this manuscript for publication.

Reviewer 2 Report

Original paper, even if retrospective. The authors highlight how the image's fusion tecniques are today particularly important to achieve the maximal surgical resection and cause minimal damage to healthy structures in close proximity to the lesion.

Author Response

Response to Reviewer #2:

Point 1. Original paper, even if retrospective. The authors highlight how the image’s fusion echniques are today particularly important to achieve the maximal surgical resection and cause minimal damage to healthy structures in close proximity to the lesion.

Response 1. We appreciate very much compliments of Reviewer #2 on our study.

Reviewer 3 Report

This is an interesting report adequately highlighting a well rounded experience with a new evolving surgical assistance technology. I totally agree that this new technology has high potential impact on improving patient safety, and the way that we conduct complex brain surgery.

Author Response

Response to Reviewer #3:

Point 1. This is an interesting report adequately highlighting a well rounded experience with a new evolving surgical assistance technology. I totally agree that this new technology has high potential impact on improving patient safety, and the way that we conduct complex brain surgery.

Response 1. We appreciate very much compliments of Reviewer #3 on our study.

Reviewer 4 Report

Authors describe an interesting retrospective experience on AR applied to the skull base meningiomas. The manuscript is well written and should be published. The only one suggestion that I would be give to the authors refers to the possibility to compare the study cases with a cases control in order to valuate if this methods really allows a surgical improvement.

Author Response

Response to Reviewer #4:

Point 1. Authors describe an interesting retrospective experience on AR applied to the skull base meningiomas. The manuscript is well written and should be published. The only one suggestion that I would be give to the authors refers to the possibility to compare the study cases with a cases control in order to valuate if this methods really allows a surgical improvement.

Response 1. We appreciate very much compliments of Reviewer #4 on our study. We thank Reviewer #4 suggestion for improvement. Lack of control group is a limitation of our study and we have pointed this out in our limitations section (Page 20, Line 517-520). There is no control group with patients who underwent resection without use of iCT-based navigation and AR. However, control group of patients who would undergo surgery without optimal conditions is considered unethical and unpractical. Main goal of our study was to demonstrate the use of AR for increased surgical precision in resection of skull base lesion. In the Discussion section we have compared our results to the results of recent studies on the subject and found comparable clinical and oncological outcomes, however literature on use of navigation and augmented reality for resection of skull base meningiomas is due to relatively recent introduction of this technique sparse. We hope that our corrections and changes are adequate and sufficient to recommend this manuscript for publication.

Round 2

Reviewer 4 Report

Should be published now